# Chromosome-Level Genome Assembly and Transcriptome Comparison Analysis of *Cephalopholis sonnerati* and Its Related Grouper Species

**DOI:** 10.3390/biology11071053

**Published:** 2022-07-13

**Authors:** Zhenzhen Xie, Dengdong Wang, Shoujia Jiang, Cheng Peng, Qing Wang, Chunren Huang, Shuisheng Li, Haoran Lin, Yong Zhang

**Affiliations:** 1College of Basic Medicine, Nanchang University, Nanchang 330031, China; xiezhenzhengirl@163.com; 2State Key Laboratory of Biocontrol, Guangdong Provincial Key Laboratory for Aquatic Economic Animals and Southern Marine Science and Engineering Guangdong Laboratory (Zhuhai), School of Life Sciences, Sun Yat-Sen University, Guangzhou 510275, China; wangdengdong@hotmail.com (D.W.); jiangshoujia@163.com (S.J.); lshuish@mail.sysu.edu.cn (S.L.); lsslhr@mail.sysu.edu.cn (H.L.); 3Guangdong Key Laboratory of Animal Conservation and Resource Utilization, Guangdong Public Laboratory of Wild Animal Conservation and Utilization, Institute of Zoology, Guangdong Academy of Sciences, Guangzhou 510260, China; pengcheng@giz.gd.cn; 4College of Marine Sciences, South China Agricultural University, Guangzhou 510642, China; wangqing@scau.edu.cn; 5Hainan Chenhai Aquatic Products Co., Ltd., Sanya 572000, China; huangchunren0704@163.com

**Keywords:** *Cephalopholis sonnerati*, pacbio sequencing, chromosome-level genome assembly, genome annotation, comparative genome analysis

## Abstract

**Simple Summary:**

*C. sonnerati* is an important marine fish species in coral reef ecosystems and has huge commercial value. This species serves as an excellent research model due to its complex social structures and behavior mechanisms. Nevertheless, owing to the lack of genomic resources, molecular genetic studies and genomic breeding remain unexplored in this species. Therefore, it is important to obtain more genome sequences of *Cephalopholis* grouper species for research on the classification, evolution, genetics, and biology of groupers. In the present study, we first assembled a high-quality, chromosome-level *C.*
*sonnerati* genome, providing a valuable genome resource for further studies of the genetic conservation, resistance breeding, and evolution of *C. sonnerati.*

**Abstract:**

The tomato hind, *Cephalopholis sonnerati*, is a bottom-dwelling coral reef fish, which is widely distributed in the Indo-Pacific and Red Sea. *C. sonnerati* also features complex social structures and behaviour mechanisms. Here, we present a high-quality, chromosome-level genome assembly for *C. sonnerati* that was derived using PacBio sequencing and Hi-C technologies. A 1043.66 Mb genome with an N50 length of 2.49 Mb was assembled, produced containing 795 contigs assembled into 24 chromosomes. Overall, 97.2% of the complete BUSCOs were identified in the genome. A total of 26,130 protein-coding genes were predicted, of which 94.26% were functionally annotated. Evolutionary analysis revealed that *C. sonnerati* diverged from its common ancestor with *E. lanceolatus* and *E. akaara* approximately 41.7 million years ago. In addition, comparative genome analyses indicated that the expanded gene families were highly enriched in the sensory system. Finally, we found the tissue-specific expression of 8108 genes. We found that these tissue-specific genes were highly enriched in the brain. In brief, the high-quality, chromosome-level reference genome will provide a valuable genome resource for studies of the genetic conservation, resistance breeding, and evolution of *C. sonnerati*.

## 1. Introduction

Groupers, the largest subfamily in the Serranidae family, comprise more than 160 species in 16 genera [1]. It has been demonstrated that approximately 47 grouper species are cultured in East and Southeast Asia [2]. These commercially important fishes possess the special characteristics of a long lifespan, large size, slow growth, and delayed reproduction [3]. Moreover, they usually inhabit coral reefs of tropical and subtropical coasts. Species of the genus Cephalopholis are the most abundant serranid in the Gulf of Aqaba (Red Sea) [4].

The tomato hind *Cephalopholis sonnerati* (Valenciennes) (Serranidae), belonging to the genus *Cephalopholis*, is a bottom-dwelling coral reef fish that is widely distributed in the Indo-Pacific and Red Sea (Figure 1). *C. sonnerati* is a protogynous hermaphrodite in life and feeds on small fish and invertebrates [4,5,6,7]. The fish is characterized by complex social structures and behavioral mechanisms. They naturally form social groups, with males and several females occupying individual territories within the male’s larger territory [7,8]. However, owing to anthropogenic activities such as overfishing and water pollution, wild populations of *C. sonnerati* have declined [9]. Previous studies of the genus *Cephalopholis* have focused on fishery management, species conservation [10], behavioral biology [4], nutritional biology [11], and phylogeography [12]. Nevertheless, owing to the lack of genomic resources, molecular genetic studies and genomic breeding remain unexplored in this species. Despite the fact that more than 270 aquatic organisms’ genome sequences have been published (https://www.ncbi.nlm.nih.gov/genome/browse#!/overview/fish (accessed on 8 October 2020)), only three genome sequences of grouper species, i.e., the giant grouper *Epinephelus lanceolatus* [13], the red-spotted grouper *Epinephelus akaara* [14] and the leopard coral grouper *Plectropomus leopardus* [15], are available. At present, no effective genome resources have been reported for the *Cephalopholis* species. Therefore, it is important to obtain more genome sequences of *Cephalopholis* grouper species for research on the classification, evolution, genetics, and biology of groupers.

A valid reference genome will provide a strong foundation for further study of the genus *Cephalopholis*. PacBio (a single-molecule real-time (SMRT) sequencing), a new third-generation sequencing technology, generates long reads with uniform coverage and high consensus accuracy compared with the second-generation sequencing technology, which generates short reads [16]. Moreover, the third-generation sequencing technology is less expensive than the second-generation sequencing technology and does not depend on amplification for library generation [17]. Additionally, Hi-C, a chromosome conformation capture-based method, can convert chromatin interactions, translating topological chromatin structures into digital information [18]. For these reasons, Hi-C has become a mainstream technology in 3D genomics.

In the present study, we reported the first high-quality, chromosome-level genome assembly of *C. sonnerati*. The genome assembly was obtained using PacBio long-read sequencing and Hi-C sequencing technologies. Our reference genome will provide a solid foundation for studies of the genetics conservation, resistance breeding and evolution of *C. sonnerati*.

## 2. Material and Methods

### 2.1. Sample and Tissue Collections

An adult female *C. sonnerati* (0.8 kg, at one year old, Figure 1) was bred at the farm of Hainan, Dongfang, Gancheng, China, was used for genome sequencing and assembly. The fish was dissected immediately after treatment with 0.2 M eugenol. Genomic DNA of *C. sonnerati* was collected from the caudal vein by a Qiagen Blood & Cell Culture DNA Midi Kit (Shanghai, China) that was used for genome sequencing. The muscle tissue was used for Hi-C library construction to obtain a chromosome-scale genome assembly. To comprehensively cover the transcripts for tissue-specific expression, tissues from the liver, gill, intestines, kidney, head kidney, brain, pituitary, gonad, heart, skin, and muscle were collected and quickly frozen in liquid nitrogen before RNA sequencing, and then the tissues were kept at −80 °C at Nanchang University.

### 2.2. Library Construction and Genome Assembly

High-quality genomic DNA was extracted from blood samples using a cetyltrime-thylammonium bromide (CTAB)-based extraction method. The quality and quantity of the extracted DNA were examined using a NanoDrop 2000 spectrophotometer (NanoDrop Technologies, Wilmington, DE, USA), a Qubit dsDNA HS Assay Kit on a Qubit 3.0 Fluorometer (Life Technologies, Carlsbad, CA, USA), and electrophoresis on 0.8% agarose gels.

A paired-end sequencing library with an insertion length of 250 bp was constructed using the VAHTS Universal DNA Library Prep Kit for MGI (Vazyme, Nanjing, China). The Agilent 2100 Bioanalyzer (Agilent Technologies, Santa Clara, CA, USA) was used to validate the purity and size distribution. Then, the obtained library was sequenced with the paired-end, 150-bp mode using the MGI-SEQ2000 platform by Frasergen Bioinformatics Co., Ltd. (Wuhan, China).

The DNA extracted from the blood was also used for sequencing library construction using the PacBio SEQUEL Platform. Ten micrograms (g) of *C. sonnerati* genomic DNA were used for 20 kb template library preparation using the BluePippin Size Selection system (Sage Science, Beverly, MA, USA) following the manufacturer’s protocol. The library was sequenced on the Pacific Biosciences Sequel II platform.

For gene annotation, transcriptome sequencing was performed with 11 tissues (liver, gill, intestines, kidney, head kidney, brain, pituitary, gonad, heart, skin, and muscle) of *C. sonnerati*. Total RNA was extracted with the TRIzol reagent (Invitrogen, Waltham, MA, USA) according to the manufacturer’s instructions. The concentration and integrity of total RNA were estimated using an Agilent 2100 Bioanalyzer (Agilent Technologies, Santa Clara, CA, USA) and ethidium bromide staining of 28S and 18S ribosomal bands on a 1% agarose gel, respectively. The verified RNA samples were equally pooled for the RNA libray construction and sequencing. In brief, the full-length cDNA was prepared using a SMARTerTM PCR cDNA Synthesis Kit (Takara Biotechnology, Dalian, China). The SMRTbell libraries were constructed with the Pacific Biosciences DNA Template Prep Kit 2.0. Library. Library quantification and size were checked using a Qubit 3.0 Fluorometer (Life Technologies, Carlsbad, CA, USA) and a 2100 Bioanalyzer system (Agilent Technologies, CA, USA), respectively. Subsequently, SMRT sequencing was conducted with the PacBio Sequel II platform by Frasergen Bioinformatics Co., Ltd. (Wuhan, China).

The short reads from the BGI platform were quality-filtered by HTQC v 1.92.310 [19] using the following method. Firstly, the adapters were removed from the sequencing reads. Second, read pairs were excluded if any one end had an average quality lower than 20. Third, the ends of reads were trimmed if the average quality was lower than 20 in the sliding window size of 5 bp. Finally, read pairs with any end shorter than 50 bp were removed. Then, the quality filtered reads were used for genome size estimation. We estimated the size of the *C. sonnerati* genome using the *K*-mer analysis per-formed with GCE [20].

The draft assembly of the genome was assembled using mecat2 [21] with default parameters. To correct errors in the primary assembly, we used gcpp 1.9.0 to polish the genome after the initial assembly was completed. In addition, we used BGI-derived short reads to correct any remaining errors by Pilon 1.22 [22]. Finally, we used BUSCO v3.0 [23] with actinoperygii_odb9 to evaluate the completeness of the assembled genome.

### 2.3. Pseudochromosome Construction

The muscle tissue of *C. sonnerati* was used for Hi-C library construction in our study. The Hi-C experiment included the following steps [18]. First, a white muscle sample of *C. sonnerati* was cross-linked using formaldehyde, and then lysed. Subsequently, chromatin was digested with MboI and proximity ligated with T4 DNA ligase. After ligation, cross-linking was reversed using 200 μg/mL proteinase K (Thermo, Shanghai, China) at 65 °C overnight. DNA purification was achieved through the QIAamp DNA Mini Kit (Qiagen) according to the manufacturer’s instructions, and the purified DNA was sheared to a length of 300–500 bp. Finally, the purified DNA was used for Hi-C library construction, and genomic DNA was sequenced on the MGI-SEQ2000 platform in 150PE mode.

The reads from the Hi-C library sequencing were mapped to the polished genome using BWA (bwa 0.7.17) with the default parameters. Paired reads that were mapped to different contigs were used to construct the Hi-C associated scaffolding. Lachesis [24] was further applied to order and orient the clustered contigs. Then, Jucier (v1.6.2) [25] was used to visually correct the assembly errors.

### 2.4. Annotation of Repeats

Two methods were combined to identify the repeat contents in the genome: homology-based and de novo prediction. For homology-based analysis, we identified the known TEs within the *C. sonnerati* genome using RepeatMasker 4.0.9 [26] with the Repbase TE library [27,28]. Repeat Protein Mask searches were also conducted using the TE protein database as a query library. For de novo prediction, we constructed a de novo repeat library of the *C. sonnerati* genome using RepeatModeler (http://www.org/RepeatModeler/ (accessed on 22 February 2020)), which can automatically execute two core de novo repeat finding programs, namely, RECON v1.08 [29] and RepeatScout (v1.0.5) [30], to comprehensively conduct, refine and classify consensus models of putative interspersed repeats for the *C. sonnerati* genome. Furthermore, we performed a de novo search for long terminal repeat (LTR) retrotransposons against the *C. sonnerati* genome sequences using LTR_FINDER (v1.0.7) [31]. We also identified tandem repeats using the Tandem Repeat Finder (TRF) package [32] and the non-interspersed repeat sequences, including low-complexity repeats, satellites and simple repeats, using Repeat Masker. Finally, we merged the library files of the two methods and used Repeat Masker to identify the repeat contents.

### 2.5. Gene Prediction and Annotation

To predict protein-coding genes in the assembled genome of *C. sonnerati*, we used three strategies: homology, de novo and transcriptome sequencing. First, protein sequences from *Epinephelus lanceolatus*, *Plectropomus leopardus*, *Epinephelus akaara*, *Oreochromis niloticus*, *Lates calcarifer*, *Gymnodraco acuticeps*, *Pseudochaenichthys georgianus* and *Cyclopterus lumpus* were downloaded from Ensembl [33] and aligned with *C. sonnerati* for homology annotation. Exonerate (v2.2.0) was used to conduct homology-based gene prediction. Second, we adopted Augustus (v3.3.1) [34] and Genescan [35] to perform de novo gene prediction. Third, protein-coding gene prediction based on transcriptome sequencing data was conducted using GMAP (version 4 July 2018) [36]. TransDecoder (3.0.1) (https://github.com/TransDecoder/TransDecoder (accessed on 10 November 2016)) was used to form the gene structure. Finally, Maker (v3.00) [37] was used to integrate the prediction results of the three methods to predict gene models. Functional annotation of the protein-coding genes was performed according to the best match of the alignments to the non-redundant (NR) (http://ftp.ncbi.nih.gov/blast/db/FASTA (accessed on 13 January 2020)), TrEMBL [38]. InterPro [39], and SwissProt [38] protein databases using BLASTP (NCBI blast v2.6.0+) [40,41] and the Kyoto Encyclopedia of Genes and Genomes (KEGG) database [42], with an e-value threshold of 1e-5. The protein domains were annotated using PfamScan (pfamscan_version) [43] and InterProScan (v5.35 74.0) [44] based on InterPro protein databases. The motifs and domains within gene models were identified by PFAM databases [45]. Gene Ontology (GO) [46] IDs for each gene were obtained from Blast2GO [47].

In addition, we used tRNAscan SE (v1.3.1) algorithms [48] and tRNAscan with default parameters to identify the genes associated with tRNA. For rRNA identification, we first downloaded the closely related species’ rRNA sequences from the Ensembl database. Then, rRNAs in the database were aligned against our genome using BlastN [40,41] with a cut-off e-value < 1 × 10^−5^, identity ≥ 85%, and match length ≥ 50bp. MiRNAs and snRNAs were identified by the Infernal (v1.1.2) [49] software against the Rfam (v14.1) database [45] with default parameters.

### 2.6. Genome Evolution Analysis

To identify the gene families for phylogenetic tree construction, we compared the genome assembly of *C. sonnerati* with those of other teleosts, including *Epinephelus lan-ceolatus*, *Plectropomus leopardus*, *Epinephelus akaara*, *Oreochromis niloticus*, *Lates calcarifer*, *Gymnodraco acuticeps*, *Pseudochaenichthys georgianus*, *Cyclopterus lumpus*, *Danio rerio*, *Salmo salar*, *Monopterus albus*, *Gadus morhua*, *Oncorhynchus mykiss*, and Oryzias latipes. *Latimeria chalumnae* was used as an outgroup. All the proteins were extracted and aligned using BLASTP [41] programs (NCBI blast v2.6.0) with a maximal e-value of 1 × 10^−5^. The OrthoFinder [50] method was used to cluster genes from these different species into gene families.

To reveal the phylogenetic relationships between *C. sonnerati* and the aforemen-tioned fishes, protein sequences from 678 single-copy orthologous gene clusters were used for phylogenetic tree reconstruction. The protein sequences of the single-copy orthologous genes were aligned with the MUSCLE (v3.8.31) [51] program, and the corresponding coding DNA sequence (CDS) alignments were generated and concatenated with the guidance of the protein alignment. RAxML (v8.2.11) [52] was used to construct the phylogenetic tree via the maximum likelihood method. The phylogenetic relationships of the other fish species were consistent with previous studies. We used the MCMCTree program of the PAML package [53] to estimate the divergence time among species. Furthermore, the 24 *C. sonnerati* chromosomes were aligned with *E. lanceolatus*, *P. leopardus* and *E. akaara* chromosomes by Mummer (v 4.0.0 beta2) [54] to identify syntenic blocks.

Based on the identified gene families and the constructed phylogenetic tree, as well as the predicted divergence time of those fish, we used CAFE [55] to analyze gene family expansion and contraction. In CAFE, a random birth and death model was proposed to study gene gain or loss in gene families across a specified phylogenetic tree. Then, a conditional *p*-value was calculated for each gene family, and a family with a conditional *p*-value less than 0.05 was considered to have an accelerated rate for gene gain or loss. These expanded and contracted gene families in *C. sonnerati* (*p*-value ≤ 0.05) were mapped to KEGG pathways for functional enrichment analysis, conducted using the enrichment methods. This method implemented hypergeometric test algorithms and the Q-value (FDR, False Discovery Rate) was calculated to adjust the p-value using the R package (https://github.com/StoreyLab/qvalue (accessed on 14 March 2019)).

Based on the phylogenetic tree, we estimated the rate ratio (ω) of nonsynonymous (Ka) to synonymous (Ks) nucleotide substitutions using the PAML (v4.9e) package [53] to examine the selective constraints on the candidate 678 single-copy orthologous genes. After the high-quality alignments of related sequences were obtained, as described above, we compared a series of evolutionary models in the likelihood framework using the species trees. A branch-site model was used to detect the average value of ω across the tree (ω0), ω of the appointed branch to test (ω2), and ω of all the other branches (ω1).

### 2.7. Identification of Differentially Expressed Genes (DEGs)

To identify the differentially expressed genes in the genome of *C. sonnerati*, 11 tis-sues (liver, gill, intestines, kidney, head kidney, brain, pituitary, gonad, heart, skin, and muscle) were used to conduct transcriptome sequencing. For each of the samples, the trimmed short reads were mapped to the genome sequence using Tophat (v2.1.1; https://ccb.jhu.edu/software/tophat (accessed on 3 April 2014)). RSEM (v1.3.0; https://deweylab.github.io/RSEM (accessed on 25 July 2017)) was used to calculate isoform level expression in terms of FPKM and TPM (transcripts per million). Only the uniquely mapped reads were used to assess gene expression. Differentially expressed genes (DEGs) between sample groups were evaluated by DESeq2 [56]. The corrected read count data of genes were imported into the R package edgeR to identify DEGs with the criteria log2FoldChang>1, a false discovery rate (FDR) and adjusted *p*-value of < 0.05, and expression (FPKM ≥ 1) in at least one sample for each comparison.

## 3. Results and Discussion

### 3.1. Genome Assembly and Assessment

A total of 57.12 Gb (54.70 × sequence coverage) clean short-reads sequencing data were obtained (Appendix A). Then, the quality clean reads were used for genome size estimation by *K*-mer analysis [20]. Accordingly, the estimated genome size of the *C. sonnerati* was 1015 Mb, with the proportion of repeat sequences and the heterozygosity rate determined to be 42.99% and 0.84%, respectively (Appendix A).

A total of 152.21 Gb ((145.8 X sequence coverage) subreads data were obtained from the SMRT bell library (Appendix A), which were used for de novo assembly of the genome. The average read length was 23.11kb, with an N50 length of 32kb (Appendix A). In total, the genome size of *C. sonnerati* was about 1043.66 Mb, with an N50 length of 2.48 Mb, accounting for 97.3% of the assembled genome size and containing 939 contigs (Table 1).

The estimation of basic genome characteristics is important for the adoption of suitable assembly strategies. The *K*-mer method is an efficient tool for estimating genome characteristics [20]. Our *K*-mer analysis showed that the estimated genome size of the *C. sonnerati* was 1015 Mb, lower than the final genome size (1043.66 Mb), based on PacBio sequencing data. This difference might be due to the collapse of repetitive elements and statistical differences in these methods. Similar results have also been reported in previ-ous research, i.e., *Triplophysa tibetana* [57], *E. akaara* [14], and Spotted Scat (*Scatophagus argus*) [58]. Moreover, the genome size of the *C. sonnerati* was shorter than the genomes of the red-spotted grouper (1135 Mb) [14] and the giant grouper (1086 Mb) [13], but longer than the genome of the leopard coral grouper (881.55 Mb) [15]. In addition, the heterozygosity of *C. sonnerati* (0.84%) was higher than that in *E. akaara* (0.375%) [14], *P. leopardus* (0.635%) [15], *M. peelii* (0.113%) [59], *S. lucioperca* (0.14%) [60], and *T. tibetana* (0.1%) [57], suggesting high population genetic diversity for the *C. sonnerati* grouper and non-selective breeding in aquaculture. Repeat content was approximately 42.99% in *C. sonnerati*, higher than that in *P. leopardus* (34.6%) [15] and *E. lanceolatus* (41.01%) [13], but lower than that in *E. akaara* (43.02%) [14]. It has been demonstrated that the repeat content and the genome in Perciformes have a strong correlation [60].

The high heterozygosity might increase the difficulty of genome sequencing and assembly in *C. sonnerati*; we obtained a high-quality chromosome-level genome using PacBio sequencing and Hi-C technology. The average read length and N50 of PacBio reads were 23.11 kb and 32 kb, respectively. The contig N50 was 2.48 Mb, higher than that of most published aquaculture species, i.e., *Glyptosternon maculatum* (993 kb) [61], sea bass (*Lateolabrax maculatus*, 31kb) [61], arapaima (*Arapaima gigas*, female: 325 kb and male: 285 kb) [62], and blunt snout bream (*Megalobrama amblycephala*, 49.4kb) [63].

### 3.2. Pseudochromosome Construction

Hi-C was applied for genome assembly. About 118.96 Gb (113.94 X sequence coverage) clean reads were used to construct the chromosome-level genome of *C. sonnerati* (Appendix A). Based on the Hi-C results, a total of 767 contigs were anchored into 24 pseudochromosomes (Chr) with a total length of 1.02Gb, with a contig N50 of 2.52 Mb and a scaffold N50 of 44.48 Mb (Table 1, Figure 2). The length of the pseudochromosomes ranged from 23.20 Mb to 50.85 Mb, containing 98.01% of the total sequence (Appendix A).

The final assembled genome was subjected to Benchmarking Universal Single-Copy Orthologs (BUSCO) with OrthoDB to evaluate the completeness of the genome assembly. Overall, 97.2% of the benchmarking universal single-copy orthologs (BUSCO) genes in the Actinoperygii Odb9 database were present as complete genes in the assembled genome, with 94.3% of BUSCO genes being complete and single copy, 2.9% being complete and duplicated, 0.8% being fragmented, and 2.0% being missed (Table 1). In addition, chromosome genome assembly is important for genome comparison and evolutionary studies [64,65]. Finally, we successfully clustered 767 contigs into 24 pseudo-chromosomes of the *C. sonnerati* genome, consistent with the number of chromosomes in other groupers [13,14,15].

### 3.3. Repetitive Element Annotation

We used de novo and homolog prediction to detect repeat sequences. Repeat sequences of 526.92 Mb in length, accounting for 50.47%, were identified in the assembled genome of the *C. sonnerati*. The TEs represented 47.23%, with 493.11 Mb of the assembly genome. Furthermore, the TEs were divided into four groups: DNA transposons (24.82%), long interspersed elements (LINEs, 13.74%), long terminal repeats (LTRs, 6.72%) and short interspersed nuclear elements (SINEs, 1.96%) (Appendix A).

### 3.4. Gene Prediction and Annotation

We annotated the protein-coding genes of the assembled genome using de novo prediction, homology-based prediction, and transcriptome sequencing-based gene prediction. A total of 26,130 protein-coding genes were obtained, with an average gene length of 20,599.55 bp. Based on de novo methods, a total of 28,361–32,602 genes were detected. Based on the homologous protein sequences of *Epinephelus lanceolatus*, *Plectropomus leopardus*, *Epinephelus akaara*, *Oreochromis niloticus*, *Lates calcarifer*, *Gymnodraco acuticeps*, *Pseudochaenichthys georgianus*, and *Cyclopterus lumpus*, a total of 41,589–53,773 proteins were obtained. Based on transcriptome sequencing data, a total of 36,352 genes were detected (Appendix A).

For the predicted non-coding genes, 373 microRNAs (miRNAs), 2232 transfer RNAs (tRNAs), 169 ribosomal RNAs (rRNAs) and 515 small nuclear RNAs (snRNAs) were also identified in the genome of *C. sonnerati* (Appendix A).

To check the quality of the annotated genes, the statistics of the predicted gene models were compared with those of eight of the closest teleost species (mentioned above). The results displayed similar distribution patterns in the exon and intron number, gene and CDS lengths, exon and intron lengths, and gene and CDS gene contents of *C. sonnerati* (Appendix A). We performed a functional annotation of all the predicted genes with InterPro, GO, KEGG, KO, SwissProt, TrEMBL and NR databases. In total, 24,629 genes (approximately 94.26%) were functionally annotated to these databases, including 21,713, 16,331, 24,250, 14,971, 22,270, 24,372, and 24,574 genes mapped to the InterPro, GO, KEGG, KO, SwissProt, TrEMBL and NR databases, respectively (Table 1). Among these, 24,574 genes (approximately 99.78%) were functionally annotated in the NR databases that had the highest annotation rate among all the above-mentioned databases.

Through de novo prediction, homology-based prediction, and transcriptome-sequencing-based gene prediction, a total of 26,130 protein-coding genes were obtained in the *C. sonnerati* genome. The number of predicted proteins in *C. sonnerati* was higher than those in *S. lucioperca* (21,249) [60] and sea bass (*Lateolabrax maculatus*, 22,015) [61], but lower than those in *M. peelii* (26,539) [59]. These differences may be due to the different statistical methods of genome annotation. These comprehensive annotated genes will facilitate future evolutionary and comparative studies within the *Cephalopholis* family.

### 3.5. Genome Evolution Analysis

To investigate the phylogenetic relationship between *C. sonnerati* with other species, a phylogenetic tree was constructed for the genomes of 15 selected species. A total of 92.35% (24,132) of the 26,130 protein-coding genes were clustered into 17,125 ortholog groups (Figure 3). The average ortholog group contained 1.41 genes per group, with 127 unique ortholog groups (Appendix A). We identified 698 single-copy orthologues using the sequencing similarities among protein-coding genes in the 15 selected species. The maximum likelihood (ML) phylogenetic tree showed that *C. sonnerati* diverged approximately 41.70 million years ago (mya) from a common ancestor with *E. lanceolatus* and *E. akaara*. In addition, other than the two Epinephelus species, *P. leopardus* was the closest sequenced relative to *C. sonnerati*, separating from their common ancestor from 66.4 to 75.7 mya (Figure 4).

Furthermore, gene family evolution was analyzed by constructing the orthologous gene families of the four groupers (*C. sonnerati*, *E. akaara*, *E. lanceolatus* and *P. leopardus*) and the results were illustrated in a Venn diagram (Appendix A). Specifically, the gene family numbers were highly similar in *C.*
*sonnerati* and *P. leopardus*, with 17,125 and 17,205 gene families, respectively. The four groupers shared 14,512 genes, of which 406 genes were specific to *C. sonnerati*. In addition, to reveal the similarities and differences between the four grouper genomes, we conducted functional comparative genomic analyses. We found that the four species had the same karyotype (2*n* = 48) with excellent genomic collinearity, revealed by the results of chromosome syntenic comparisons. *C. sonnerati* chromosomes (Ch) 1, 2, 21, 22, 23, and 24 displayed strong synteny with those of Ch 1, 2, 21, 22, 23, and 24 in the *P. leopardus* genome, indicating that Ch 1, 2, 21, 22, 23, and 24 likely separated from a common ancestor (Figure 5).

The characteristics of the expansion and contraction of gene families play significant roles in the research of phenotypic diversity and environmental adaptation [66]. A total of 1224 expanded gene families and 1977 contracted gene families were identified in the *C. sonnerati* genome in comparison with 15 closet species (Figure 4). The expanded gene families were mainly associated with sensory system. Additionally, 112 positively selected genes were found in the *C. sonnerati* genome (Appendix A). Next, 18 KEGG pathways and 33 GO terms were significantly enriched from the expanded gene families. Some expanded genes related to “binding” and “cellular process” were enriched in the GO enrichment analysis (Appendix A). Interestingly, the KEGG enrichment analysis revealed that the expanded gene families were highly enriched in the sensory system, including “olfactory transduction” and “taste transduction” (Appendix A). These functional analyses of expanded genes suggested that these genes might play important roles in the sensory system development of *C. sonnerati*, such as in the taste and lateral-line systems. It has been demonstrated that the adaptability of the taste system in fish has contributed to differences in the life history, feeding ecology, and morphology of fish [67].

The sensory system of vertebrates is determined by both genetic and environmental factors. Thus, the key genes related to the sensory system of *C. sonnerati* were analyzed. In vertebrates, teneurins (Ten-m/Odz) are a phylogenetically conserved family of type II transmembrane proteins encoded by four genes (teneurin 1–4) and expressed in interconnected regions of the nervous system [68]. In the present study, the teneurin gene families were included in the enriched GO category. The teneurin-2 (Tenm2) and teneurin-3 (Tenm3) genes were significantly highly expressed in the sensory system, suggesting their functioning in the vision of *C. sonnerati* (Appendix A). Additionally, the chemokine-like receptor 1 (CMKLR1) was enriched in the sensory system. Previous studies have shown that CMKLR1 is also expressed in the brains of humans, mice, and rats [69]. Our results suggested that CMKLR1 plays an important role in the development of the fish brain; further study is required concerning the functional mechanism of this gene in *C. sonnerati*.

### 3.6. Identification of Differentially Expressed Genes (DEGs)

To identify the differentially expressed genes in the genome of *C. sonnerati*, 11 tissues (liver, gill, intestines, kidney, head kidney, brain, pituitary, gonad, heart, skin, and muscle) were used to conduct the transcriptome sequencing. A total of 8,108 tissue-specific expression genes were found in the *C. sonnerati* genome based on the edgeR methods. Next, we employed the KEGG pathway database to perform functional annotation for all these genes (Appendix A). Functional analyses of these pathways indicated that they included complement and coagulation cascades-related genes, DNA replication-related genes, synaptic vesicle cycle-related genes and long-term potentiation-related genes. Finally, we compared the number of DEGs in 11 tissues and found that the tissue-specific genes were highly enriched in the brain of *C. sonnerati* (Figure 6). Interestingly, we found that most genes related to signal transduction and the nervous system were specifically expressed in the brain (Appendix A).

We compared the number of DEGs in 11 tissues and found that the tissue-specific genes were highly enriched in the brain of *C. sonnerati* (Appendix A). Interestingly, we found that most genes related to signal transduction and the nervous system are specific to the brain (Appendix A). For example, calcium voltage-gated channel auxiliary subunit gamma 5 (CACAG5), protein kinase C beta (PRKCB), and fibroblast growth factor 13 (FGF13) were enriched in the MAPK signaling pathway; ionotropic glutamate receptor (NMDA 1), alpha-1B adrenergic receptor, and 5-hydroxytryptamine receptor 7 were enriched in the calcium signaling pathway; 5-hydroxytryptamine 6 receptor, calcium/calmodulin dependent protein kinase IV, and AP-1-like transcription factor were enriched in the cAMP signaling pathway (Appendix A). Additionally, protein kinase C alpha (PKC-α), 5-hydroxytryptamine receptor 1F, and 5-hydroxytryptamine receptor 1D were enriched in the nervous system (Appendix A).

## 4. Conclusions

In this study, we obtained the first high-quality, chromosome-level genome of *C. sonnerati* using a combination of BGI short reads and PacBio and Hi-C sequencing technologies. The genome size was about 1043.66 Mb, with an N50 length of 2.49 Mb and a scaffold N50 of 44.48 Mb. A total of 26,130 protein-coding genes were predicted in the assembled genome. Moreover, 1224 expanded gene families and 1977 contracted gene families were identified in the assembled genome in comparison to the 15 closest species. *C. sonnerati* diverged approximately 41.7 million years ago (mya) from a common ancestor with *E. lanceolatus* and *E. akaara*. We found tissue-specific expression for 8108 genes. The reference genome obtained in this study will greatly improve our understanding of the genetic diversity of serranids and promote the development of comparative evolutionary research.

## Figures and Tables

**Figure 1 biology-11-01053-f001:**
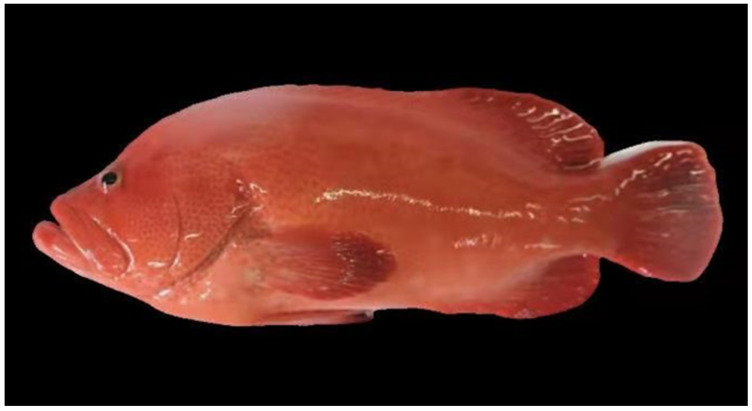
A picture of *C. sonnerati* for genome sequencing.

**Figure 2 biology-11-01053-f002:**
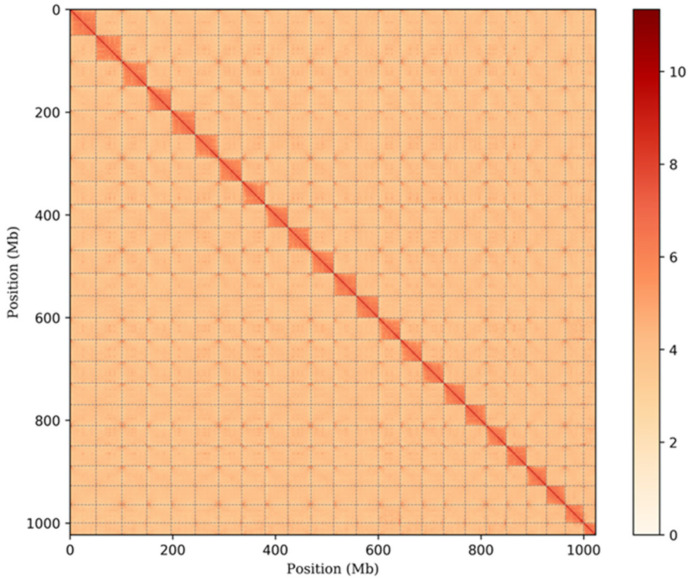
The *C. sonnerati* genome contig contact matrix using Hi-C data. The color bar indicates contact density from red (**high**) to white (**low**).

**Figure 3 biology-11-01053-f003:**
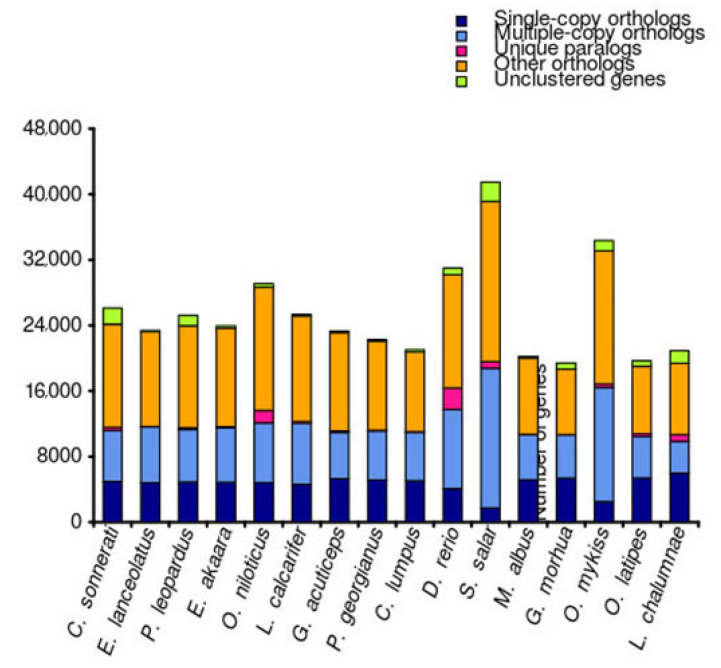
Orthologous genes between *C. sonnerati* and other 15 teleost species.

**Figure 4 biology-11-01053-f004:**
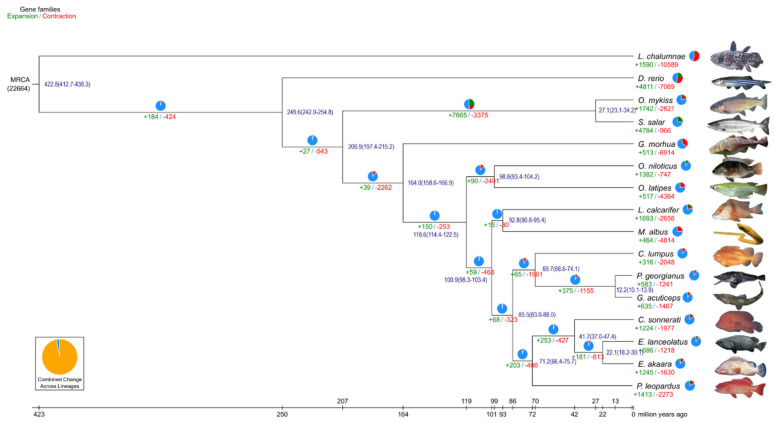
*C. sonnerati* diverged from other species and their phylogeny. The blue numbers are the divergence time of the prediction. The numbers below the branches are the numbers of expanded and contracted gene families (green, expanded; red, extracted). The scale at the bottom represents divergence time, and the one-time unit represents 100 million years ago. The pie charts represent gene families (black, expanded; red, extracted; blue, others).

**Figure 5 biology-11-01053-f005:**
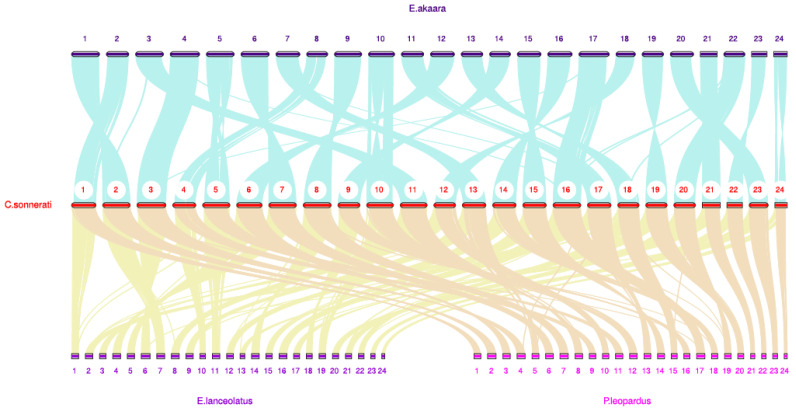
Chromosome synteny analysis between *C. sonnerati* and other three groupers. LG 1–24 represents chromosomes 1–24 of *C. sonnerati* and other three groupers, respectively.

**Figure 6 biology-11-01053-f006:**
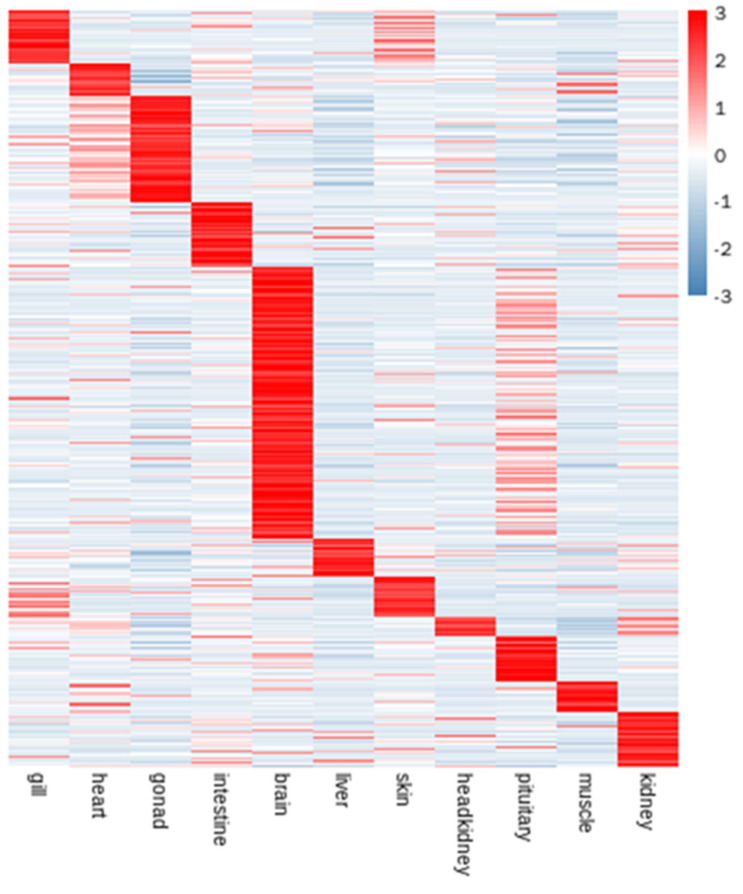
The heatmap of tissue-specific expression genes in 11 tissues of *C. sonnerati* genome. Genes with a tissue specificity score absolute value of 1 were considered to show tissue specificity.

**Table 1 biology-11-01053-t001:** Summary of the *C. sonnerati* Genome Assembly and Annotation.

Chromosome-Level Genome Assembly	
Genome assembly and chromosomes construction	
Contig N50 size (bp)	2,482,587
Contig N90 size (bp)	683,704
Maximum contig size (bp)	12,345,001
Total contigs number	939
Total length of genome	1,043,655,803
Number of chromosomes (bp)	24
Total length of chromosomes (bp)	1,022,871,484
Scaffold N50 (bp)	44,482,143
Contig N50 (bp)	2,517,244
Final Assembly Genome Quality Evaluation	
Proportion of complete BUSCOs (%)	97.2
Proportion of complete and single-copy BUSCOs (%)	94.3
Proportion of complete and duplicated BUSCOs (%)	2.9
Proportion of fragmented BUSCOs (%)	0.8
Proportion of missing BUSCOs (%)	2.0
Gene Annotation	21,173
Number of InterPro annotation	16,331
Number of GO annotation	24,250
Number of KEGG annotation	14,971
Number of KO annotation	22,270
Number of SwissProt annotation	24,372
Number of TrEMBL annotation	24,574
Number of NR annotation	24,629
Number of all annotation	1501
Unannotated	

## Data Availability

All data supporting the funding of this study has been deposited in the NCBI Sequence Read Archive under the BioProject number PRJNA699118. All of the raw sequencing data, including BGI and PacBio Sequel II reads, are also deposited under the same BioProject number. The transcriptome data of C. sonnerati are deposited in the NCBI Sequence Read Archive under the number SUB9299153 and SUB9901994.

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
