# Peer review of "Chromosome-Level Genome Assembly and Transcriptome Comparison Analysis of Cephalopholis sonnerati and Its Related Grouper Species"

_biology, 2022, doi:10.3390/biology11071053_

Round 1

Reviewer 1 Report

First of all, I'd like to thank all the authors of this manuscript for delivering this piece of scientific work. Authors are very dedicated to this work and spend significantly of time and money to accomplish this.

However, academic English writing might challenge authors a bit. For example, the title is a little bit awkward and might need a new one. Please consider changing it to "Chromosome-level genome assembly and transcriptome comparison analysis for Cephalopholis sonnerati and its related grouper species". For other editing advice, I've provided detailed minor revision suggestions in the attached word file.

The presentation of the figure would be better if the authors considering add a few more details. More specific suggestions could be found in the attached word file.

Overall, this is a standard whole-genome scale investigation for an economically valuable fish species. Although there is nothing quite fancy in the results and discussion, the value of filling the gap of knowledge is still pretty high. I'd recommend editor to accept the manuscript with minor revision required.

Reviewer 2 Report

Authors constructed the first chromosome-level genome of C. sonnerati based on the short reads ,PacBio and Hi-C sequencing technology. The results provided the useful reference genome. Some minor revision should be made.

Line 108:“that were” should be changed into “were”.

Line 165-167: “Interestingly, the KEGG enrichment analysis revealed that the expanded gene families were highly enriched in the sensory system, including “olfactory transduction” and “taste transduction”. “olfactory transduction” and “taste transduction” are usually related to the feeding habits.Authors should add more information.

Figure 4 and 5 are vague, Please change good quality figure.

Line 228-230: “The number of predicted proteins in C. sonnerati was higher than those in E. akaara (23,808), P. leopardus (25,248), and E. lanceolatus (24,718). These differences may be due to different statistical methods in genome annotation.” It is incorrect. The number differences must be found in different species. Please rephrase it.

Reviewer 3 Report

The study by Zhang et al. combination used PacBio, Illumina, and Hi-C technologies to assemble a Cephalopholis sonnerati chromosome-level reference genome with a size of 1043.66 Mb (N50: 2.48Mb), Hi-C loading rate of 94.26%, and 26,130 protein-coding genes. Comparative genome analyses indicated that the expanded genes families were highly enriched in the sensory system. Overall,I think the topic is a worthy one, and the analyses are thorough. This study will provide valuable resources for future breeding. I agree to publish this article with a little correction.

Minor comments:

1.    Please check the diagram and the legend.

2.    Please check the format of article, and make it meet the requirement of the journal.

3.    Line 275-277 “Our phylogenetic analysis revealed that C. sonnerati diverged approximately 41.7 million years ago (mya) from the common ancestor with E. lanceolatus and E. akaara.” Replace 41.7 for 41.70. Please check the number of decimal places in the article for consistency.

Reviewer 4 Report

In this paper, the authors sequenced, assembled and analyzed the genome of the tomato hind, Cephalopholis sonnerati. They used computational methods to annotate the genome and compared the genome to those of other fish, particular three other species of grouper.  They also used RNA-seq data from eleven different tissues to identify genes with tissue-specific expression and then performed GO term and pathway enrichment analysis with those genes.

The experiments described in this paper are scientifically sound and the results indicate that the authors have produced a high-quality genome assembly that will be extremely helpful to researchers studying C. sonnerati and related fish. However, there are some problems with the presentation and interpretation of the results as detailed below. 

Line 21 AND lines 91-92 

The report of BUSCO percentages needs to be reworded.  It currently sounds like complete genes, complete single-copy genes and complete duplicated genes are different categories of genes in the BUSCO database, rather than characterizations of the BUSCO genes in this specific genome.  I would suggest something like “Overall, 95.8% of the benchmarking universal single-copy orthologs (BUSCO) genes in the Actinoperygii Odb9 database are present as complete genes in the assembled genome, with 91.7% of BUSCO genes being complete and single copy and 4.1% being complete and duplicated.”

It also seems that BUSCO analysis was only done on the assembled genome before Hi-C scaffolding was performed.  The BUSCO analysis should be repeated on the final assembly and results added to Table 1.

Line 87 – The assembled genome is LARGER than the K-mer analysis, so 97.3% is not correct as stated. Instead, the genome size estimated by K-mer analysis is 97.3% of the assembled genome size.

Line 102 – According to Table S4, the smallest pseudochromosome is 23.2 Mb not 2.32 Mb as stated in the text.

Line 147 and line 235

In two places, the authors state that P. leopardus was the most closely related species to C. sonnerati in the phylogenetic tree.  However, E. akaara and E. lanceolatus are definitely closer to C. sonnerati.  It needs to be reworded to say something like “other than the two Epinephelus species, P. leopardus is the closest sequenced relative”.

Line 149-152 – The description of what is shown in the Venn diagram (Figure S3) is not clear.  What type of genome comparison was used to produce the diagram? Also, why would a similar number of gene families between two species indicate “highly conserved synteny”? 

Lines 155-158 – I don’t think the conclusions drawn from the C. sonnerati and P. leopardus synteny are correct.  Too much significance is placed on some of the syntenic chromosomes having the same chromosome number. Since the chromosomes are numbered according to size, it just means that those chromosomes happen to fall in the same size position in the two species.  Addition or loss of repetitive sequences could easily change the lengths of chromosomes without much effect on synteny of genes.  The chromosomes that show conserved synteny with different numbered chromosomes were still present in the common ancestor.  Their length has just changed relative to some of the other chromosomes in at least one of the species.  I also don’t understand why the authors invoke chromosome fusion as an explanation for the synteny between C. sonnerati and P. leopardus.  Since all of the grouper species examined have the same number of chromosomes and there is basically one-to-one synteny of chromosomes between all the species, it doesn’t seem that chromosome fusion has been an important factor in the evolution of grouper species.

Line 180 – How is tissue-specific expression being defined?  Do the criteria described in the Methods section (lines 436-437) for DEGs have to be met for comparisons with all the other tissues? Meaning that expression is at least 2 fold higher in one tissue than in every other tissue? In Figure 6 it looks like there is some expression of the tissue-specific genes in other tissues, so it would be helpful to explain the criteria more clearly.

Lines 188-189

The conclusion here seems to be incorrect. I think the authors only looked at the predicted functions of the tissue specific genes, so they can’t conclude that most genes related to signal transduction and the nervous system are specific to the brain.  Maybe they meant that many of the genes specifically expressed in the brain were related to signal transduction and the nervous system.  Or perhaps that most of the TISSUE-SPECIFIC genes related to signal transduction and the nervous system were expressed in the brain.

Line 213

The word “size” is missing after “genome”.

Line 324 

I happened to notice that the reference for GCE is incorrect here.  It should be reference 19.  This seems to be an isolated problem as it is correct earlier in the text and other reference numbers I spot checked are correct.

Discussion section – Much of the Discussion section is material that should really be in the Results section.  The journal format allows combination of the Results and Discussion sections, so I would recommend that as a way to correct this issue.

Figure Legends – Both in-text and supplementary figures need more detailed legends describing what is shown in the figures.  Several figures have almost no legend and are very difficult to interpret.
